# Hematopoietic Stem Cell Transplantation in Acute Promyelocytic Leukemia in the Era of All-Trans Retinoic Acid (ATRA) and Arsenic Trioxide (ATO)

**DOI:** 10.3390/cancers15164111

**Published:** 2023-08-15

**Authors:** Andrei Colita, Alina Daniela Tanase, Ciprian Tomuleasa, Anca Colita

**Affiliations:** 1Department of Hematology, Carol Davila University of Medicine and Pharmacy, 020021 Bucharest, Romania; 2Department of Hematology, Coltea Clinical Hospital, 030171 Bucharest, Romania; 3Department of Bone Marrow Transplantation, Fundeni Clinical Institute, 022338 Bucharest, Romania; 4Department of Transplant Immunology, Carol Davila University of Medicine and Pharmacy, 020021 Bucharest, Romania; 5Department of Hematology, Research Center for Functional Genomics and Translational Medicine, Iuliu Hatieganu University of Medicine and Pharmacy, 400012 Cluj Napoca, Romania; 6Department of Hematology, Ion Chiricuta Clinical Cancer Center, 400015 Cluj Napoca, Romania; 7Department of Pediatrics, Carol Davila University of Medicine and Pharmacy, 020021 Bucharest, Romania

**Keywords:** acute promyelocytic leukemia, relapse, hematopoietic stem cell transplantation

## Abstract

**Simple Summary:**

Acute promyelocytic leukemia (APL) currently benefits from first-line treatment based on all-trans retinoic acid and arsenic trioxide, ensuring long-term complete responses for most patients. However, a proportion of 5–20% of patients relapse, and their long-term survival, even in the context of therapy with the mentioned drugs, is no longer as favorable, requiring the association of a highly efficient consolidation. Current recommendations indicate hematopoietic stem cell transplantation as consolidation treatment in patients with relapsed APL who achieve a new complete remission. Our article aims to present the current data on the role of transplantation in APL and the aspects regarding this therapy that are still under debate, as well as new data on possible alternatives to this type of treatment.

**Abstract:**

Acute promyelocytic leukemia (APL) currently represents one of the malignant hemopathies with the best therapeutic responses, following the introduction of all-trans retinoic acid (ATRA) and subsequently of arsenic trioxide (ATO) treatment. As a result, a large proportion of patients with APL achieve long-term responses after first-line therapy, so performing a hematopoietic stem cell transplant as consolidation of first complete remission (CR) is no longer necessary. Even in the case of relapses, most patients obtain a new remission as a result of therapy with ATO and ATRA, but an effective consolidation treatment is necessary to maintain it. The experience accumulated from studies published in the last two decades shows the effectiveness of hematopoietic stem cell transplantation (HSCT) in improving the outcome of patients who achieve a new CR. Thus, the expert groups recommend transplantation as consolidation therapy in patients with a second CR, with the indication for autologous HSCT in cases with molecular CR and for allogeneic HSCT in patients with the persistence of minimal residual disease (MRD) or with early relapse. However, there is a variety of controversial aspects related to the role of HSCT in APL, ranging from the fact that outcome data are obtained almost exclusively from retrospective studies and historical analyses to questions related to the type of transplantation, the impact of minimal residual disease, conditioning regimens, or the role of other therapeutic options. All these questions justify the need for controlled prospective studies in the following years.

## 1. Introduction

Acute promyelocytic leukemia (APL) is a distinct subtype of acute myeloid leukemia (AML), representing 10–15% of newly diagnosed AML cases [1,2,3]. At the onset, it is characterized by abnormal white blood cell (WBC) counts with distinctive blast morphology, thrombocytopenia, coagulopathy, and a tendency to severe bleeding, which makes APL a medical emergency requiring prompt diagnosis and treatment [2,4]. APL is cytogenetically characterized by the presence of a balanced translocation that involves the retinoic acid receptor alpha (RARA) gene on chromosome 17 (17q21). In the vast majority of APL cases, t(15;17) (q22;q12–21) is present, leading to the fusion of the promyelocytic leukemia (PML) gene on chromosome 15 with the RARA gene, thus resulting in the formation of the PML–RARA fusion transcript. In rare cases, rearrangements of 17q21 lead to the fusion of RARA to alternative partner genes such as NPM (nucleophosmin) associated with t(5;17)(q35;q12–21), ZBTB16 (zinc finger and BTB domain-containing protein 16) (former PLZF—promyelocytic leukemia zinc finger) with t(11;17)(q23;q21), NuMA (nuclear mitotic apparatus) t(11;17)(q13;q21) and other variants (14 variants known to date) [5]. The resulting RARA fusion disrupts normal RARA signaling, leading to a blockage in the differentiation and maturation of myeloid cells, causing an accumulation of immature promyelocytes in the bone marrow and peripheral blood [2,6,7,8].

The introduction over the last decades of differentiating agents targeting the specific genetic aberration involved in pathogenesis, such as all-trans retinoic acid (ATRA) and arsenic trioxide (ATO), has turned APL into the most curable acute leukemia with almost 90% long-term survivors [2,4,9]. However, 5–20% of APL patients relapse after achieving initial remission [9,10]. The use of ATO in the therapy of relapse has improved outcomes due to its effects of tumor cell growth inhibition and induction of apoptosis reported in APL cells, but also in other malignant hemopathies as well as in malignant solid tumors [11,12]. Still, the second relapse rate (RR) remains high (approximately 41–48%) [13,14,15], raising the issue of the most effective post-remission treatment with the aim of maintaining a new remission for as long as possible. The role of HSCT in APL has been a subject of study over the last decades and has changed over time due to the development and emergence of new therapies. Thus, from extensive use in frontline therapy in the pre-ATRA era, it currently has extremely limited indications in first complete remission (CR) in the context of very effective ATRA and ATO-based therapies, while the main recommendation has shifted to the post-remission treatment in relapsed patients. Currently, salvage therapy in relapsed APL consists of induction of a second CR, followed by consolidation with HSCT. Due to better tolerability and lower treatment-related toxicity, autologous HSCT (auto-HSCT) is preferred in CR2, while allogeneic HSCT (allo-HSCT) is reserved for patients relapsing after less than 1 year, failing to achieve molecular CR2 or for those in second or further relapse [4,9,10]. Although HSCT’s place in managing these patients is defined by current guidelines, there are still various controversial aspects regarding its role in APL.

In the present article, we aim to provide an overview of the currently available data on HSCT in APL patients by discussing aspects related to indications, type of transplant, stem cell source, donor selection, conditioning regimens, prognostic factors, the impact of minimal residual disease, and future perspectives.

## 2. Transplantation in Newly Diagnosed APL

According to the European Leukemia Net (ELN) guidelines, therapy for newly diagnosed patients consists of induction, consolidation, and maintenance treatment guided by risk stratification (high-risk disease—WBC count >10 × 10^9^/L, non-high-risk disease—WBC count ≤ 10 × 10^9^/L respectively) [16]. Induction therapy consists of combinations of ATRA with ATO, ATRA with anthracycline-based chemotherapy, or ATRA with ATO and chemotherapy, depending on the risk stratification. Moreover, consolidation and maintenance are carried out in accordance with the category of risk (for non-high-risk patients—chemotherapy-free consolidation and no maintenance, while for high-risk patients, consolidation with ATRA + chemotherapy is followed by maintenance) [16]. This therapeutic strategy leads to high CR rates of over 90%, even 100% in some studies, along with high event-free survival (EFS) rates (80–97%) and overall survival (OS) rates (93–99%) [2,17,18,19].

These very good results of first-line therapy led to a decrease in the importance of HSCT in APL in CR1. Indeed, since the early era of ATRA therapy when HSCT was still largely used in CR1, the number of HSCT procedures has registered a steady decrease in CR1 APL patients due to the improvement of induction results and the introduction of ATO [20,21,22,23]. Currently, the consensus expressed by the most recent recommendations and guidelines of ELN, the European Society for Blood and Marrow Transplantation (EBMT), and the National Comprehensive Cancer Network (NCCN) is that HSCT is not recommended in CR1 in APL patients, regardless of their risk stratification [16,24,25].

Although recent outcome data show that disease resistance has practically disappeared in almost all patients with genetically proven PML/RARA APL, CR being achieved with current therapies, there are still reports of rare cases with molecular persistence of disease at the end of consolidation that require immediate additional treatment, including HSCT if feasible [16].

A special situation is that of APL variants (APLv), resistance or at least a low sensitivity to differentiation-inducing agents being reported in 4 of the 14 variants described so far: poor response of ZBTB16-RARA and STAT5b-RARA variants to both ATRA and ATO, 1 case of BCoR-RARA insensitive to ATO, and 1 case of TBLR1-RARA variant insensitive to ATRA [16,26,27,28,29]. The outcome of these variants is associated with relapse rates of 30–80% reported in several studies [30,31]. A recent study presented 24 cases of APLv followed over 12 years in several centers around the world, including 18 cases with ZBTB16-RARA, 3 STAT5B-RARA, and 1 each with PRKAR1A-RARA, NuMA-RARA, and FIP1L1-RARA rearrangements [5]. In this series, there was no evidence of differentiation with ATRA and/or ATO in patients with either ZBTB16-RARA or STAT5B-RARA rearrangements, and most of these patients received transplant-based treatment [5].

## 3. Relapsed APL

Despite the advances in APL therapy through the introduction of differentiation-inducing agents, there is still a proportion of 5–20% of cases that relapse [9,10,13]. In the great majority of cases, relapse occurs during the first three years [2,32]. The incidence of relapses is lower in patients initially treated with ATRA + ATO than in those treated with ATRA + chemotherapy [17,18,19]. Relapse is defined as hematological relapse by the reappearance of >5% abnormal promyelocytes in the bone marrow, or molecular relapse defined as two successive PCR-positive assays, with stable or rising PML-RARA transcript levels detected in independent bone marrow samples analyzed in two laboratories [10,22]. Early initiation of therapy, immediately after the detection of molecular relapse, provides a better outcome than treatment after the appearance of frank hematological relapse [33,34], hence the recommendations to initiate preemptive therapy [16,25,35].

The molecular mechanisms leading to relapse are incompletely described. They may involve resistance to ATRA as a result of increased catabolism and decreased delivery of ATRA to cell nucleus, mutations in the ligand-binding domain of the RARA portion of the fusion protein, occurrence of additional mutations (e.g., Flt3) or additional chromosomal abnormalities, as well as mutations in the PML domain with decreased sensitivity to ATO. These mechanisms might differ depending on the initial therapy and the time of relapse [36,37,38,39]. In a study conducted on 45 patients with relapse/progression after ATRA + chemotherapy initial therapy, mutations in the ligand binding domain of PML-RARA were found in 18 cases (40%), of which 11 cases with progression during ATRA therapy and 7 with relapse after the end of ATRA therapy [36]. Another study of 35 patients with relapse after ATO identified point mutations within the B2 domain of PML in 9 patients with ATO-resistant disease, while 7 patients with ATO-resistant disease simultaneously presented RARA mutations. In this study, patients having only RARA mutations responded to ATRA + ATO therapy, while none with both PML and RARA mutations responded [39].

In accordance with this data, current guidelines provide that therapy for molecular or hematologic relapse should be chosen considering the previously used first-line treatment. However, the proposed therapies differ between the recommendations of various expert groups. Thus, the ELN guideline recommends a “cross-therapy” scheme: APL patients relapsing after ATRA + chemotherapy should be treated with an ATRA + ATO-based approach as salvage therapy until the achievement of minimal residual disease (MRD) negative status based on RT-PCR, whereas those relapsing after ATO-based therapy should receive ATRA + chemotherapy. Patients with late relapse (CR1 duration > 2 years) could be exempted from crossing over to a different treatment of relapse [16]. On the other hand, in the NCCN guideline, the cutoff between early and late relapses is set at 6 months. Patients with early relapse after initial therapy with ATRA + ATO will be treated with regimens based on anthracyclines or gemtuzumab-ozogamicin (GO) +ATRA, while those with initial treatment with ATRA + chemotherapy will receive ATO ± ATRA ± GO. Patients with late relapse after first-line therapy with ATO will be treated at relapse with ATO ± ATRA ± chemotherapy or GO [25]. Studies analyzing treatment with ATO in relapsed APL cases following initial anthracycline-based treatment showed high CR2 rates of 80–85% or higher [13,15]. There is less data regarding treatment outcomes with ATRA + chemotherapy in cases with relapse after ATRA + ATO initial therapy, with molecular CR reported rates of 91% and 0.74 probability of disease-free survival (DFS) at 4 years [40]; therefore, the treatment recommendation was based on an expert consensus rather than on data from clinical experience [41]. However, recent data showed that in younger APL patients relapsing after frontline ATO, reinduction with an ATO-based regimen could be effective, regardless of the time elapsed from the first complete remission [41]. The second RR is also relatively high, even after ATO-based therapy (approximately 41–48%) [2,15]. Thus, the main objective of therapy in relapsed APL is the achievement of molecular CR as a bridge to an efficient consolidation therapy [10,16].

The choice of the appropriate post-remission therapy in CR2 is dependent on factors specific to the patient (age, comorbidities, donor availability) or disease-specific variables (depth of response-molecular status at the end of treatment, duration of first remission, presence of mutations).

At present, most expert groups suggest HSCT is the best consolidation therapy after salvage therapy for relapsed APL. Both ELN and NCCN guidelines recommend auto-HSCT as the first choice for eligible patients achieving second molecular remission, while patients failing to achieve molecular remission should undergo an allo-HSCT [16,25]. The eighth report of the EBMT covering the indications for hematopoietic cell transplantation recommends autologous and allogeneic transplantation from matched sibling donors as the standard of care for APL in molecular CR2 [24].

These recommendations derive from published data on managing relapsed APL, suggesting a better survival after consolidation with HSCT. These available data regarding the role of auto- and allo-HSCT following relapse in APL are mainly from retrospective single-center studies, uncontrolled comparisons, and registry data analyses, reporting outcomes with different types of induction and salvage therapies, conditioning regimens, and various follow-up periods [13,15,41,42,43,44,45,46] (Table 1). Thus, the earliest information comes from studies analyzing the outcome with or without HSCT in relapsed APL patients treated with ATRA + chemotherapy. A retrospective analysis of the European Acute Promyelocytic Leukemia Group published in 2005 performed on 122 patients with CR2 after ATRA + chemotherapy showed a superior relapse-free survival (RFS), EFS, and OS at 7 years in the autologous HSCT (auto-HSCT) group (79.4%, 60.6%, and 59.8% respectively) and allogeneic HSCT (allo-HSCT) group (92.3%, 52.2%, and 51.8% respectively) compared to patients not receiving transplantation (38%, 30.4%, and 39.5%, respectively) [42]. A retrospective analysis of the JALSG APL97 study reported the outcome of 57 APL patients with CR2 after ATRA-based therapy in APL97. The 5-year EFS rate, OS rate, and cumulative incidence of relapse (CIR) were 50.7%, 77.4%, and 51.0% in the non-HSCT group (30 patients), 41.7%, 83.3%, and 58.3% in the auto-HSCT group (6 patients) and 71.1%, 76.2%, and 9.8% in the allo-HSCT group (21 patients), respectively [43]. Subsequently, several studies comparing evolution with and without HSCT in patients with relapsed APL who received salvage therapy with ATO were published [13,15,41,44,46,47].

A retrospective ELN study analyzing the outcome of 155 patients treated with ATO in the first relapse showed a favorable significant prognostic impact of auto- and allo-HSCT on OS and leukemia-free survival (LFS) compared to patients without HSCT [15]. A retrospective study on data from the Center for International Blood and Marrow Transplant Research (CIBMTR) and EBMT registries analyzed 207 patients with relapsed APL receiving ATO and compared the outcome of 67 patients receiving ATO alone and 140 with auto-HSCT. At 5 years, OS was 42% and 78% for the ATO-only and auto-HSCT groups, respectively (*p* < 0.001) [46].

As data from published studies and expert groups recommendations indicate a better outcome of relapsed APL patients with HSCT consolidation, there are still some issues under debate, mainly due to insufficient or conflicting data and the lack of prospective, controlled studies: influence of pre-transplant therapy, type of transplant, stem cell source, best type and timing of mobilization and stem cell harvesting for auto-HSCT, donor type, conditioning regimen, influence of minimal residual disease status prior to HSCT, prognostic factors, possible alternatives to HSCT.

### 3.1. Pre-Transplant Salvage Therapy

A limited amount of information, with conflicting results, is available regarding the impact of the type of pre-transplant therapy on patient outcomes. A retrospective multicentric study on 58 patients undergoing autologous HSCT for APL at 21 institutions in the United States (U.S.) and Japan reported significantly longer times to neutrophil recovery (median 12 days vs. 9 days, *p* < 0.001) as well as lower median viable post-thaw CD34 + cell recovery of cryopreserved autologous stem cell products from patients with prior treatment with ATO, suggesting that ATO exposure prior to CD34 + cell harvest has deleterious effects on hematopoietic recovery after auto-HSCT. In this study, relapse-free survival (RFS) and OS were shorter in patients receiving ATO prior to stem cell collection [48]. Douer et al. analyzed data from 2 U.S. studies that enrolled 45 relapsed patients treated with ATO, of which 18 underwent HSCT (4 cases with auto-HSCT, 14 with allo-HSCT) after completing ATO therapy. At the end of follow-up, all patients receiving auto-HSCT were alive without relapse, while in the allo-HSCT group 71% of patients were alive, indicating that ATO therapy is associated with lower toxicity and appears to reduce the risk of transplantation-related complications [47]. In an analysis of data from CIBMTR/EBMT registries, in the auto-HCT group, 79 patients received ATO as part of the salvage therapy before transplant, and 54 received ATRA + chemotherapy, with similar OS of the two groups (*p* = 0.274) [49]. In a large retrospective study of the CIBMTR data on 62 patients with auto-HSCT and 232 patients with allo-HSCT from 79 centers in 18 countries between 1995–2006, there was no impact of ATO pre-transplant therapy on the risk of relapse after HSCT evaluated by univariate and multivariate analysis [46]. Another retrospective registry-based study from the Japanese Transplant Registry Unified Management Program involved 198 patients with APL who underwent auto-HSCT during the second CR2 from 1995 to 2012. Patients transplanted after ATO-based therapy had significantly better RFS, OS, and CIR compared to those treated without ATO, whereas non-relapse mortality (NRM) did not differ between the two groups, suggesting that the introduction of ATO may result in significant improvements in overall outcomes for relapsed APL patients undergoing auto-HSCT during CR2 [50]. Although ATO therapy is well tolerated by most patients, certain pharmacological aspects require caution in its pre- and post-transplantation use. ATO therapy can prolong the QT/QTc interval, which can lead to a higher risk of arrhythmias in the peri-transplantation period, in which transient cardiac abnormalities and electrolyte imbalances may appear. Hence, the recommendation is to restrict ATO therapy in the 30 days preceding and following HSCT [47].

Another aspect is the lack of consensus regarding the best therapy prior to the HSCT event in the ATO era. In the above-mentioned Japanese study, patients received three courses of ATO therapy prior to the peripheral blood stem cells (PBSC) collection [50]. In the ELN study, induction ATO monotherapy was followed by a second ATO or ATRA + ATO course as consolidation [15]. Other authors used ATO monotherapy until CR2 was achieved, followed by at least two consolidation cycles with ATO [13]. No available data so far can indicate whether two courses are sufficient or if more therapy is needed prior to HSCT, or whether MRD negativity needs to be achieved in the bone marrow [51].

### 3.2. Type of Transplant

An important aspect of conducting efficient consolidation therapy after obtaining CR2 is choosing the appropriate type of transplant. Generally, auto-HSCT is associated with a better safety profile-lower NRM, absence of graft-versus-host disease (GVHD), and better quality of life (QoL)-counterbalanced by a higher risk of relapse, while the performance of allo-HSCT is followed by a lower RR but at the price of higher NRM and deterioration of QoL caused by treatment complications including GVHD [51,52]. Comparative data of auto-vs. allo-HSCT in the setting of CR2 APL patients are available from a reduced number of retrospective uncontrolled studies and registry data analyses (Table 2) [13,15,20,22,42,43,45,49,53,54,55]. Most published data provide support in favor of auto-HSCT mainly due to better results in terms of OS [13,42,45,51,53] and NRM [22,49,54,55], even if some studies revealed significantly higher RR [13,43] when compared to allogeneic transplant. In the largest study published so far, data from the EBMT Registry, including 228 patients with allo-HSCT and 341 patients with auto-HSCT, were analyzed in terms of LFS, OS, RR, and NRM. The 2-year probabilities of LFS, OS, RR, and NRM were 75%, 82%, 23%, and 3% for auto-HSCT and 55%, 64%, 28%, and 17% for allogeneic HSCT, respectively. In the multivariate analysis, LFS, OS, and NRM were better for patients undergoing auto-HSCT than for those undergoing allo-HSCT [55].

In addition to data from retrospective studies, evidence of the outstanding efficacy and feasibility of auto-HSCT after induction and consolidation with ATO in relapsed APL was provided by a prospective study on 23 patients who underwent auto-HSCT, demonstrating 5-year EFS, OS, and NRM rates of 65%, 77%, and 0%, respectively [56].

Currently, auto-HSCT is widely accepted as the preferred treatment for patients with relapsed APL who have obtained CR, particularly when the patients are in molecular CR [16,25].

### 3.3. Stem Cell Source in Auto-HSCT: Stem Cell Harvesting

Most studies on auto-HSCT reported peripheral blood as the main stem cell source [14,15,23,41,42,43,44,45,46,49,50,54,55,56]. Peripheral blood stem cells (PBSC) were harvested after obtaining remission and assessed by morphological and karyotypic examination [42] or, in most cases, after molecular examination and achievement of MRD negativity [41,42,43,56]. Mobilization of PBSC was performed with agranulocyte colony-stimulating factor (G-CSF) (variable dose—5 or 10 µg/kg/day) in a steady state or after chemotherapy (various regimens) [41,42,43,44,56]. There are practically no data regarding comparative results on mobilization therapy, except for occasional observations in a few studies: in the Japanese phase 2 prospective trial on auto-HSCT, PBSC harvest after high-dose cytarabine chemotherapy is presented as part of the sequential treatment of relapsed APL that lead to high efficacy in this setting [56]; in an Indian study including 35 patients, with PBSC collection after mobilization with G-CSF (10 µg/kg/day for 4 days) following ATO-based induction therapy, 37.1% of patients required a second-day harvest to achieve target stem cell dose [41]. An alternative option to overcome poor PBSC mobilization in relapsed APL patients after repetitive chemotherapy and ATO therapy could be using a CXCR4 blockade with plerixafor. Mobilization with plerixafor is rarely and cautiously used in AML due to the concern of mobilizing leukemia cells. Data published in several studies shows CXCR4 expression on leukemic blasts, increased CXCR4 expression on APL cells following ATRA therapy, and the demonstration of increased time-dependent mobilization of APL blasts after CXCR4 blockade in a murine model [57,58,59,60]. Data regarding the clinical use of plerixafor in APL are scarce and come from 2 isolated case reports showing that mobilization with plerixafor in combination with G-CSF, with or without cyclophosphamide, led to a safe collection of PML-RARA PCR negative grafts [61,62].

There are few data on the impact of stem cell sources in auto-HSCT, and the very few studies that analyzed the influence of stem cell sources on outcomes showed no statistical differences [22,23]. Yet, there are some observations regarding the outcome difference after auto-HSCT of bone marrow stem cells or PBSC in MRD-positive cases. An earlier Italian prospective study enrolled 7 patients with detectable MRD before auto-HSCT, and all 7 experienced early relapses after performing bone marrow stem cell transplantation [63]. Data from an observational study from the CIBMTR included 6 patients with positive MRD before auto-HSCT who presented similar DFS and OS with 35 patients with negative MRD [48]. A large retrospective study of the Japanese Society for Hematopoietic Cell Transplantation (JSHCT) reported 35 patients with positive MRD and 293 MRD-negative before performing auto-HSCT using mainly PBSC, with no association between MRD status and transplant-related mortality (TRM), relapse, and OS rates [23]. Another recent analysis of data collected from the Japanese Society for Transplantation and Cellular Therapy (JSTCT) and the Japanese Data Center for Hematopoietic Cell Transplantation between 2006 and 2019 on 296 patients with PBSC auto-HSCT included 21 cases with detectable MRD. In this study, MRD-positive status had no significant impact on outcome [64]. Some authors speculated that PBSC auto-HSCT could be feasible in patients with PML-RARA positive bone marrow due to several mechanisms: effective eradication of residual disease in vivo, preferential mobilization into the autograft of short-term repopulating cells, but too few leukemia stem cells to induce relapse, the non-clonogenic nature of the PML/RARA-positive cells present in the graft or a modest purging effect of cryopreservation on unstable leukemic clones [20,23,48,64,65]. These aspects should be interpreted with great caution due to the small number of cases, retrospective collection of PML-RARA results, and lack of assay standardization [23,48,64]. However, current guidelines recommend auto-HSCT as the first choice for patients achieving second molecular remission, while patients failing to achieve molecular remission should undergo an allo-HSCT [16,25].

The results of allo-HSCT, despite providing a strong antileukemic effect through pre-transplantation conditioning therapy and the post-transplantation immunologic graft-versus-leukemia (GVL) effect, are undermined by the association of complications impacting QoL and especially by the increased risk of NRM [64]. In the context of APL patients in CR2 with better-reported outcomes after auto-HSCT, the high toxicity associated with allo-HSCT is less acceptable, recommending this procedure for selected patients with reduced benefit from auto-HSCT, such as those who cannot achieve CR, failing to achieve molecular CR and/or relapsing after auto-HSCT [10,16,25,66]. Data regarding the results of allo-HSCT are provided by a small number of studies, very few of which have enrolled significant numbers of patients (Table 2). A survey of the EBMT activity in APL patients between 1993–2003 analyzed 137 patients receiving allo-HSCT in CR2 with 5-year LFS, RR, and TRM of 59%, 17%, and 24%, respectively [22]. Data reported to CIBMTR from 1995 to 2006 included 232 patients receiving allo-HSCT in CR2 with results inferior to auto-HSCT (5-year EFS, OS, RR, and TRM of 50%, 54%, 18%, and 31%, respectively) [49]. An analysis of EBMT transplant activity between 2004 and 2018 reported 228 allo-HSCT patients with an inferior 2-year OS compared to auto-HSCT [55]. A large retrospective analysis of Japanese nationwide transplantation registry data of patients with relapsed APL receiving allo-HSCT exclusively between 2006 and 2020 reported 195 patients, including 69 who underwent transplantation in non-CR and 55 who relapsed after prior auto-HSCT, with a median duration of follow-up of 5.4 years. The 5-year OS rates for patients with allo-HSCT in CR and non-CR were 58% and 39%, respectively, if they did not receive a prior auto-HSCT. In the patients relapsing after an auto-HSCT, the 5-year OS rate was 47% for those with allo-HSCT in CR and 6% for those transplanted without achieving CR (*p* = 0.001). The study concluded that allo-HSCT is effective in selected, relapsed APL patients with less expected benefit after auto-HSCT. A particularly dismal outcome is reported for patients who relapsed after auto-HSCT failing to obtain further CR [66].

In the setting of allo-HSCT, the stem cell source varied. Some groups used mainly bone marrow (proportions ranging from 64% to 87%) [22,42,43,49,53], while others used PBSC in most patients (53–79% of patients) [45,55,67]. Only one study reported that the use of mobilized PBSC was associated with decreased TRM in patients receiving allo-HSCT in CR2 (*p* = 0.008) [22]. There were no other differences in outcome associated with stem cell sources reported by other studies.

### 3.4. Donor Type

Most studies on allo-HSCT in APL patients reported the use of stem cells from matched sibling donors (MSD) in a much higher proportion than in other types of leukemia, with percentages varying between 38–100% [20,22,42,43,44,45,49,53,54,55,67]. Some earlier studies even established the achievement of allo-HSCT from MSD as an inclusion criterion [22,53]. The proportion of unrelated donors and alternative donors is higher in more recent studies, probably reflecting the current trends and advances in survival following the use of these donor types [49,55,66]. The EBMT registry data analysis included donor type 130 MSD (57%), 83 matched and mismatched unrelated donors (36%), 4 haploidentical donors (2%), 5 cord blood units (2%), and 6 cases receiving stem cells from other relatives (2%) [55]. In the study from the Japanese nationwide transplantation registry, donor sources included 48 related donors (HLA-matched in 39), 89 unrelated donors (HLA-matched in 68), and 58 single-unit umbilical cord blood (all HLA-mismatched). There were no significant differences regarding OS, relapse, or NRM related to donor type [66].

### 3.5. Conditioning Treatment

In the setting of auto-HSCT, most studies reported the exclusive use of myeloablative conditioning (MAC) regimens [15,43,44,45,46,50,53,56]. Only 2 studies analyzing the data from the CIBMTR Registry and the EBMT Registry reported 8% and, respectively, 14% of cases receiving reduced intensity conditioning (RIC) [55,65]. Chemotherapy-based conditioning was preferred in most studies on auto-HSCT [22,23,44,45,50,53,54,55,56,64]. Most groups preferred BU/CY as conditioning therapy [22,44,45,53,54,55], except for some Japanese studies in which BU/MEL was more frequently used [23,50,56,64]. Moreover, the Japanese authors reported some correlations between the conditioning regimens used and the outcome of patients with auto-HSCT. An analysis of the 25-year experience (1992–2006) of the JSHCT on auto-HSCT in APL patients showed that conditioning with BU/MEL had a protective effect against relapse (*p* = 0.018) [23]. Another study of prognostic factors in a series of 296 patients with APL performing auto-HSCT during second or subsequent complete remission (CR2+) between 2006 and 2019 showed that conditioning regimens not including busulfan were significantly associated with a shorter RFS (univariate analysis) and higher risk of NRM (multivariate analysis) [64]. Three studies reported preferential TBI-based conditioning for auto-HSCT [42,43,49]. A retrospective analysis of the European APL Group reported twice as many relapses in patients receiving CY-TBI conditioning compared to those who received BU/CY (*p* = 0.45). It concluded that BU/CY conditioning regimen was at least as effective as the CY-TBI conditioning regimen, suggesting that TBI might be avoided in the case of auto-HSCT in APL [42]. In the allo-HSCT setting, the use of TBI was preferred in the European APL Group analysis [42], the Japan Adult Leukemia Study Group APL97 report [43], and another Japanese study [53], while non-TBI-based conditioning was preferred in the most recent EBMT Registry report [55] and other studies [45,54]. The equal use of TBI- and non-TBI-based conditioning was reported by the CIBMTR Registry [49] and the 2007 EBMT data report [22]. For non-TBI conditioning, MAC chemotherapy regimens have been used in most patients (range 68–100%) [15,43,45,49,53,55,67]. RIC has been increasingly used in recent series, reaching 25% in the Japanese nationwide transplantation registry study [66] and 32% in the latest EBMT Registry report [55]. No significant comparative data between conditioning regimens have been reported so far.

### 3.6. The Influence of MRD Status

The influence of MRD status prior to HSCT on outcome in the setting of patients with APL in CR2 was reported by several studies with conflicting results. Initial results on MRD impact were reported by Meloni et al. in a prospective study that enrolled 7 patients with detectable MRD before auto-HSCT, followed by early relapses in all cases after using bone marrow cells for transplantation [63]. Several large retrospective studies, including ELN and EBMT registries data analysis and a multicentric Italian study, showed a significant impact of pre-transplant MRD status on outcome after HSCT, especially in the setting of allo-HSCT (Table 3) [15,55,67]. Other studies showed no influence of pre-HSCT MRD positive status on relapse, treatment failure, or survival in allo-HSCT and, surprisingly, also not in auto-HSCT [23,43,49,50,64]. It is worth mentioning that these latter studies used PBSC as compared to the study of Meloni et al., in which the stem cell source was represented by bone marrow, bringing into discussion the possibility of performing PBSC auto-HSCT in patients with MRD positivity. The possible mechanisms explaining these differences are mentioned above. Nevertheless, current guidelines recommend auto-HSCT as the first choice for eligible patients achieving second molecular remission, while patients failing to achieve molecular remission should undergo an allo-HSCT [16,25].

### 3.7. Prognostic Factors

The analysis of factors influencing outcomes in the HSCT setting was reported by several retrospective studies (Table 3) [15,22,23,41,43,46,46,49,55,64,67]. The prognostic factors identified by most studies were CR1 duration [15,41,46,49,64], time from diagnosis to transplant [20,52], age [21,41,46,52], and order of remission (CR2 vs. CR ≥ 3) [64,67].

### 3.8. Consolidation Therapies as Alternative Possibilities to HSCT

Although consolidation with HSCT shows improved outcomes, it appears that a significant subset of patients could experience long-term survival with ATO-based post-remission therapy [9]. An interesting data set comes from studies that enrolled patients with relapsed APL who obtained a second molecular CR, who were offered auto-HSCT but could not continue with this procedure for non-medical reasons, usually because of financial constraints or patient choice, and who received ATO-based consolidation. An Indian study included 63 patients with CR2 and negative MRD, of which 28 opted against auto-HSCT and received ATO-based maintenance therapy for 10 days/month for 6 months. For these patients, OS and EFS (58.6% and 47.4%, respectively, at 5 years) were inferior compared to the auto-HSCT group, but this study showed that ATO maintenance could generate long-term survival for some patients [41]. A single-center Korean study showed no significant differences in survival outcomes between 19 patients receiving ATO-based post-CR therapy, 12 patients with auto-HSCT, and 6 with allo-HSCT, suggesting that ATO-based post-remission therapy is effective in patients achieving molecular CR2 [13]. A report of the NCRI AML Working Group on the long-term follow-up of the AML17 trial included 31 APL patients achieving molecular CR2, of which 18 were treated with ATO + ATRA alone without transplant or consolidation chemotherapy, and 14 remained in molecular remission after a 5-year follow-up [68].

Other options investigated by several studies are single agent GO [69], tamibarotene alone or in combination with ATO [70,71], venetoclax combinations [72,73,74], or a combination of ATO, ATRA, mitoxantrone, and bortezomib [75], as well as the use of oral ATO [76,77]. Prospective controlled studies are necessary to address the role of these various therapeutic options.

The limitations of the clinical use of HSCT in APL are related to factors specific to the patient (age, comorbidities, donor availability), disease-specific variables (MRD status before HSCT, CR1 duration), therapy-specific factors (toxicity, impact on QoL) and other non-medical issues (financial/administrative constraints, patients’ choice) [9,41,78]. High-dose conditioning chemotherapy for auto-HSCT is associated, in rare cases, with the risk of secondary malignancies (myelodysplastic syndrome, AML) [9,10,42,47]. This raises the question of whether some patients in CR2 may be consolidated without transplantation, especially those with a family history suggestive of myeloid neoplasms (which should be tested for germline predisposition mutations) [9,10]. In the setting of allo-HSCT, limitations are associated with increased risks of NRM, deterioration of QoL due to treatment and donor availability (solved in part by the growing use of alternative donors) [20,51]. From this perspective, allo-HSCT should be used for selected patients who cannot achieve CR and have relapsed after auto-HSCT [51].

## 4. Conclusions and Future Perspectives

Due to the very good results obtained with ATO and/or ATRA therapy of newly diagnosed APL cases, HSCT is no longer indicated in frontline treatment, excepting extremely rare cases of persistent MRD or APLv. At present, the main indication for HSCT is in relapsed APL as consolidation after obtaining the second remission. Current guidelines recommend autologous transplantation for patients who achieve a second molecular remission. In contrast, allogeneic transplantation should be used in cases with persistent MRD after salvage therapy, in those with short CR1, or in relapses after autologous HSCT. Controlled prospective studies are needed in the future to clarify the existing controversies.

## Figures and Tables

**Table 1 cancers-15-04111-t001:** Post-remission therapy in relapsed APL—comparison of transplantation vs. non-transplantation.

Study	Study Period/ Type	Relapse Therapy	Post- Remission Treatment	No.	RFS	EFS	OS	RR
de Botton et al., 2005 [42]	1992–2001 Retrospective Multicentric	ATRA + CT	Auto Allo Non-HSCT	50 23 49	79% (7y) 92% (7y) 38% (7y)	61% (7y) 52% (7y) 30% (7y)	60% (7y) 52% (7y) 40% (7y)	
Thirugnanam et al., 2009 [44]	1998–2006 Retrospective Unicentric	ATO—based	Auto Non-HSCT	14 19		83% (5y) 34% (5y)	100% (5y) 39% (5y)	7% (7y) 63% (7y)
Pemmaraju et al., 2013 [45]	1980–2010 Retrospective Unicentric	Various	Auto Allo Non-HSCT	10 17 16		69% (7y) 41% (7y) NA	86% (7y) 49% (7y) 40% (7y)	
Fujita et al., 2013 [45]	1997–2002 Retrospective Multicentric	ATRA + CT	Auto Allo Non-HSCT	6 21 30		42% (5y) 71% (5y) 45% (5y)	83% (5y) 76% (5y) 75% (5y)	58% (5y) 10% (5y) 51% (95y)
Lengfelder et al., 2015 [15]	2003–2011 Retrospective ELN Registry	ATO—based	Auto Allo Non-HSCT	60 33 55			77% (3y) 79% (3y) 59% (3y)	37% (3y) 39% (3y) 59% (3y)
Ganzel et al., 2016 [46]	2000–2011 Retrospective Registry data	ATO or CT based	Auto Non-HSCT	140 67			78% (5y) 42% (5y)	
Fouzia et al., 2021 [41]	1998–2015 Retrospective Unicentric	ATO—based	Auto Non-HSCT	35 28		87% (5y) 48% (5y)	90% (5y) 59% (5y)	
Min et al., 2022 [13]	2000–2019 Retrospective Unicentric	ATO or CT based	Auto Allo Non-HSCT	12 6 19	66% (3y) * 50% (3y) 44% (3y)		75% (3y) 66% (3y) 65% (3y)	41% (3y) 0% (3y) 50% (3y)
Douer et al., 2003 [47]	1997–2000 Retrospective analysis of 2 studies	ATO	Auto Allo Non-HSCT	4 14 27			4 ** 12 ** 11 **	0 ^†^ 1 ^†^ NA

** number of patients alive at the end of follow-up (May 2022); ^†^—Number of patients relapsed at the end of follow-up. CT—chemotherapy; ATO—arsenic trioxide; RFS—relapse-free survival; EFS—event-free survival; OS—overall survival; RR—relapse rate; HSCT—hematopoietic stem cell transplantation; * DFS—disease-free survival; NA—not available.

**Table 2 cancers-15-04111-t002:** Comparative results of autologous vs. allogeneic HSCT in APL patients in CR2.

Study	Study Period/Type	Relapse Therapy	HSCT Type	No.	RFS	EFS	OS	RR	NRM
de Botton et al., 2005 [42]	1992–2001 Retrospective Multicentric	ATRA + CT	Auto Allo	50 23	79% (7y) 92% (7y)	61% (7y) 52% (7y)	60% (7y) 52% (7y)		
Pemmaraju et al., 2013 [45]	1980–2010 Retrospective Unicentric	Various	Auto Allo	10 17		69% (7y) 41% (7y)	86% (7y) 49% (7y)		
Fujita et al., 2013 [43]	1997–2002 Retrospective Multicentric	ATRA + CT	Auto Allo	6 21		42% (5y) 71% (5y)	83% (5y) 76% (5y)	58% (5y) 10% (5y)	
Lengfelder et al., 2015 [15]	2003–2011 Retrospective Multicentric	ATO—based	Auto Allo	60 33			77% (3y) 79% (3y)	37% (3y) 39% (3y)	
Min et al., 2022 [13]	2000–2019 Retrospective Unicentric	ATO or CT based	Auto Allo	12 6	66% (3y) * 50% (3y)		75% (3y) 66% (3y)	41% (3y) 0% (3y)	
Kohno et al., 2008 [53]	1999–2004 Retrospective Multicentric	Various	Auto Allo	15 13	69% (4y) 46% (4y)		76% (4y) 46% (4y)	21% (4y) 9% (4y)	
Holter Chakrabarty et al., 2014 [49]	1995–2006 Retrospective Registry data	non-ATO/ ATO based	Auto Allo	62 232	63% (5y) 50% (5y)		75% (5y) 54% (5y)	30% (5y) 18% (5y)	7% (5y) 31% (5y)
Alimoghaddam et al., 2011 [54]	1989–2011		Auto Allo	11 29		52% (5y) 62% (5y)	47% (5y) 66% (5y)		0% (5y) 21% (5y)
Sanz et al., 2007 [22]	1993–2003	ATRA + CT	Auto Allo	195 137	51% (5y) 59% (5y)			37% (5y) 17% (5y)	16% (5y) 24% (5y)
Sanz et al., 2021 [55]	2004–2018 Retrospective Registry data		Auto Allo	341 228		75% (2y) 55% (2y)	82% (2y) 64% (2y)	23% (2y) 28% (2y)	3% (2y) 17% (2y)

CT—chemotherapy; ATO—arsenic trioxide; RFS—relapse-free survival; EFS- event-free survival; OS—overall survival; RR—relapse rate; HSCT—hematopoietic stem cell transplantation; * DFS—disease-free survival.

**Table 3 cancers-15-04111-t003:** Prognostic factors and influence of MRD on post-transplant outcome in APL relapsed patients.

Study	Study Period/Type	HSCT Type	Factors Influencing Outcome	Data on MRD Status/Impact on Outcome
de Botton et al., 2005 [42]	1992–2001 Retrospective Multicentric	Auto Allo		Auto-HSCT—Superior 7y RFS, EFS, OS in patients with mCR compared to patients lacking molecular analysis (*p* = NS)
Ramadan et al., 2012 [67]	2000–2010 Retrospective Multicentric	Allo	CR2 vs. CR3+ for OS (*p* = 0.05)	mCR prior to allo-HSCT—better OS (*p* = 0.03), lower CIR (*p* = 0.3)
Ganzel et al., 2016 [46]	2000−2011 Retrospective Registry data	Auto	CR1 duration for OS (*p* = 0.001), DFS (*p* = 0.002), multivariate (*p* < 0.001) Extramedullary disease—on OS (*p* = 0.046), NS in multivariate analysis	
Fujita et al., 2013 [43]	1997–2002 Retrospective Multicentric	Auto Allo	age at CR2 ≥ 40 years (*p* = 0.006)	Auto-HSCT—pre-transplant MRD had no predictive significance with respect to relapse
Lengfelder et al., 2015 [15]	2003–2011 Retrospective Multicentric	Auto Allo	CR1 duration ≥ 1.5 years (*p* = 0.006) No negative impact of extramedullary disease on transplant outcomes	mCR2 before HSCT (*p* < 0.001) (univariable and multivariable analysis)
Holter Chakrabarty et al., 2014 [49]	1995–2006 Retrospective Registry data	Auto Allo	age > 40 years for DFS (*p* = 0.005), OS (*p* < 0.001) CR1 < 12 months on OS (*p* = 0.021)	No influence of pre-HSCT MRD positive status on relapse, treatment failure, or survival in auto- and allo-HSCT
Sanz et al., 2007 [22]	1993–2003 Retrospective Registry data	Auto Allo	Auto-HSCT Year of HSCT for LFS (*p* = 0.05) Interval from diagnosis to HSCT > 18 months for LFS (*p* = 0.0001), TRM (*p* = 0.0016) Allo-HSCT Year of HSCT for RI (*p* = 0.0004), TRM (*p* = 0.03) WBC at diagnosis for RI (*p* = 0.008) Source of HSC for TRM (*p* = 0.008)	
Sanz et al., 2021 [55]	2004–2018 Retrospective Registry data	Auto Allo	Age (*p* = 0.002) Time diagnosis to HSCT (*p* = 0.006)	negative MRD before allo-HSCT~better 2y OS (*p* = 0.001), 2y LFS (*p* = 0.002)
Fouzia et al., 2021 [41]	1998–2015 Retrospective Unicentric	Auto	CR1 duration (*p* = 0.025)	
Yanada et al., 2020 [23]	1992–2016 Retrospective Multicentric	Auto	HSCT period for RI (*p* = 0.014) Age ≥ 50 years for NRM (*p* = 0.007) Male vs. female for NRM (*p* = 0.009)	No association between MRD status and TRM, relapse, and OS rates
Yanada et al., 2022 [64]	2006–2019 Retrospective Registry data	Auto	CR1 duration ≥ 2 years for RFS (*p* = 0.002) CR3+ vs. CR2 for NRM (*p* = 0.036)	MRD status–not predictive for survival outcomes
Yanada et al., 2017 [50]	1995–2012 Retrospective Multicentric	Auto	PS 0 vs. ≥1	pre-transplantation PML-RARA status—not predictive for outcomes

CR—complete remission; mCR—molecular complete remission; RFS—relapse-free survival; EFS—event-free survival; LFS—leukemia-free survival; OS—overall survival; RR—relapse rate; HSCT—hematopoietic stem cell transplantation; CIR—cumulative incidence of relapse; TRM—transplant-related mortality; RI- relapse incidence; NRM—non-relapse mortality; MRD—minimal residual disease; WBC—white blood cells; PS—performance status; NS—non-significant.

## Data Availability

Not applicable.

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
