# Peer review of "Hematopoietic Stem Cell Transplantation in Acute Promyelocytic Leukemia in the Era of All-Trans Retinoic Acid (ATRA) and Arsenic Trioxide (ATO)"

_cancers, 2023, doi:10.3390/cancers15164111_

Round 1
Reviewer 1 Report
The manuscript is well written and clearly organized,
it is easy to read and very well sourced
I have two minor comments
Concerning the low level of CD34 for auto-HSCT on APL after CR1 can the authors precise if anti-CXCR4 was used in more recent studies and discuss its potential use to increase the quality of PBSC graft.
regarding the allo HSCT, can the authors precise wether the conditionning regimens were MAC or RIC in case of non-TBI based regimen.
Author Response
Thank you for your appreciation and comments.
1. Concerning the low level of CD34 for auto-HSCT on APL after CR1 can the authors precise if anti-CXCR4 was used in more recent studies and discuss its potential use to increase the quality of PBSC graft.
Thank you for your feedback. We included the suggested data.
2. regarding the allo HSCT, can the authors precise wether the conditionning regimens were MAC or RIC in case of non-TBI based regimen.
Thank you for your feedback. We included the suggested data.
Reviewer 2 Report
This article aimed to provide an overview of the currently available data on HSCT in APL patients by discussing indications, type of transplant, stem cell source, donor selection, conditioning regimens, prognostic factors, the impact of minimal residual disease, and future perspectives.
1. The title is similar to research articles and it is better to indicate a review article.
2. It is suggested to use the following related articles in this article:
DOI: 10.7314/apjcp.2015.16.13.5191.
DOI: 10.1186/s12977-020-0513-y
DOI: 10.1634/theoncologist.8-2-132
3. It is better to write the full name of the abbreviations in the title for better understanding.
4. The limitations of the clinical use of the Hematopoietic stem cell transplant method in acute promyelocytic leukemia should be mentioned.
5. It is recommended to add a schematic figure to show the generality of the article.
6. English writing must be checked completely in terms of grammar.
7. Full names of abbreviations should be mentioned only in the first mention.
8. The introduction is short and less is written about the main topic of the article.
English writing must be checked completely in terms of grammar.
Author Response
This article aimed to provide an overview of the currently available data on HSCT in APL patients by discussing indications, type of transplant, stem cell source, donor selection, conditioning regimens, prognostic factors, the impact of minimal residual disease, and future perspectives.
- The title is similar to research articles and it is better to indicate a review article and
- It is better to write the full name of the abbreviations in the title for better understanding.
Thank you for your feedback. We have reworded the title.
- It is suggested to use the following related articles in this article:
DOI: 10.7314/apjcp.2015.16.13.5191.
DOI: 10.1186/s12977-020-0513-y
DOI: 10.1634/theoncologist.8-2-132
Thank you for your feedback. We included the information from the suggested articles in the manuscript (ref. 11, 12,47)
- The limitations of the clinical use of the Hematopoietic stem cell transplant method in acute promyelocytic leukemia should be mentioned.
Thank you for your suggestion. We added the limitations of the clinical use of hematopoietic stem cell transplant in APL as suggested.
- It is recommended to add a schematic figure to show the generality of the article.
Thank you for your feedback. We have developed a graphical summary that will be uploaded with the revised manuscript.
- English writing must be checked completely in terms of grammar.
Thank you for your feedback. The manuscript has been completely checked in terms of grammar by a colleague fluent in English writing.
- Full names of abbreviations should be mentioned only in the first mention.
Thank you for your feedback. We made the changes asked for in the paper.
- The introduction is short and less is written about the main topic of the article.
Thank you for your suggestion. We completed the Introduction section
Reviewer 3 Report
The authors described hematopoietic stem cell transplantation in acute promyelocytic leukemia in the ATRA/ATO era. This field has not little progress, but important for discussion for better therapeutic selection. This manuscript is well written, and nothing I'm concerned about.
Author Response
Thank you for your appreciation and comments.
Round 2
Reviewer 2 Report
The authors addressed all of my comments.
Moderate editing of English language is required.